# Novel Method for Monitoring Mining Subsidence Featuring Co-Registration of UAV LiDAR Data and Photogrammetry

Jibo Liu [1,*], Xiaoyu Liu [2], Xieyu Lv [3], Bo Wang [4] and Xugang Lian [3]

1 College of Mining Engineering, Guizhou University of Engineering Science, Bijie 551700, China
2 School of Geology Engineering and Geomatics, Chang'an University, Xi'an 710054, China
3 College of Mining Engineering, Taiyuan University of Technology, Taiyuan 030024, China
4 Inspection and Testing Center of Shanxi Province (Institute of Standard Metrology of Shanxi Province), Taiyuan 030012, China
* Correspondence: liujibo@gues.edu.cn

**Abstract:** Addressing the problem that traditional methods cannot reliably monitor surface subsidence in coal mining, a novel method has been developed for monitoring subsidence in mining areas using time series unmanned aerial vehicle (UAV) photogrammetry in combination with LiDAR. A dynamic subsidence basin based on the differential digital elevation model (DEM) was constructed and accuracy of the proposed method was verified, with the uncertainty of the DEM of difference (DoD) being quantified via co-registration of a dense matching point cloud of the time series UAV data. The root mean square error calculated for the monitoring points on the subsidence DEM was typically between 0.2 m and 0.3 m with a minimum of 0.17 m. The relative error between the maximum subsidence value of the extracted profile line on the main section after fitting and the measured maximum subsidence value was not more than 20%, and the minimum value was 0.7%. The accuracy of the UAV based method was at the decimeter level, and high accuracy in monitoring the maximum subsidence value was attained, confirming that an innovative strategy for monitoring mining subsidence was realized.

**Keywords:** subsidence monitoring; UAV photogrammetry; airborne LiDAR; co-registration; dynamic subsidence basin

## 1. Introduction

Monitoring of surface subsidence plays a vital role in protecting the ecological environment of mining areas and ensuring the sustainable development of modern coal mines [1]. Traditional observation technologies, such as the total work station, have some limitations, such as low efficiency, high labor costs, and an inability to produce surface subsidence basins. With the development of UAV platform and computer vision technologies, the use of structure from motion (SfM) and Multiview-Stereo (MVS) algorithms to process UAV images are increasingly being used to produce terrain data of high resolution, namely, point cloud, digital surface model (DSM), and digital orthophoto maps (DOM) for research and applications in the earth sciences [2,3].

Compared with methods based on satellite or airborne LiDAR, the dynamic, safe, low-cost, and efficient data acquisition afforded by a UAV mean that the UAV based technique has great potential for monitoring subsidence caused by coal mining. In determining the ground displacement using a UAV, research has, to date, focused on assessing its effectiveness in monitoring subsidence. Most studies have compared the DEM of two different measurement periods [4–7]. Zheng et al. [8] and Dawei et al. [9] constructed dynamic surface subsidence basins based on DEM of difference (DoD) and retrieved mining subsidence-related parameters. Miao et al., optimized the parameters of the filtering algorithm of the progressive triangulation densification filtering, and determined the subsidence area and the maximum subsidence value using airborne LiDAR [10]. Yu et al., constructed the DoD

based on the fuzzy inference system, and obtained the surface deformation information of the study area more accurately by using Bayesian estimation based on the weight filter window [11]. Lu et al., calculated the DoD, extracted surface subsidence information by setting the elevation difference and area threshold, and adopted the method based on elevation difference analysis [12]. The research of Stupar et al., showed that UAV can obtain high-density DEM, which can be used for other activities during and after mine excavation, and has good consistency with GNSS RTK data [13].

There are also studies that use dense point clouds derived from UAVs and algorithms that allow point cloud to point cloud comparison. Pal et al., used UAV photogrammetry to obtain multi-temporal point cloud data, and quantified the subsidence of the two periods through the nearest neighbor point cloud to cloud (C2C) comparison method [14]. Esposito et al., carried out multi-temporal point cloud comparison experiments based on UAV photogrammetry in open-pit mines, and verified the effectiveness of a Multiscale Model to Model Cloud Comparison (M3C2) algorithm for accurate change detection [15]. Puniach et al., compared and analyzed the effectiveness of different image registration algorithms in multi-temporal DOM matching, and proposed a workflow to automatically determine the horizontal displacement caused by underground mining [16]. Dawei et al., combined UAV photogrammetry with interferometric synthetic aperture radar (InSAR) technology to monitor ground mining subsidence basins and obtained relatively reliable mining subsidence parameters [17]. Tong et al., presented a practical framework for the integration of UAV based photogrammetry and terrestrial laser scanning (TLS) with application to open-pit mine areas, showing that the accuracy of geo-positioning based on UAV imagery can be improved [18].

UAV technology is highly appropriate for monitoring surface subsidence in coal mining. Generally, the monitoring accuracy is at the decimeter level, and a few studies have reported a centimeter capability [19,20]. Given that the level of accuracy reported in independent studies differ, it is desirable to use a variety of sensors, including LiDAR, to obtain multi-source monitoring data to facilitate the development of UAVs for multi-source monitoring [21]. When comparing multi-temporal measurements in different studies, it is essential to ensure the consistency between measurements from the different sources.

In the present study, time series UAV photogrammetry was combined with LiDAR to exploit the advantages afforded by the two different technologies in monitoring subsidence. The airborne LiDAR data was used as a reference to co-register the multi-temporal UAV photometry data, which improved the repeatability of UAV multi-temporal data. Co-registration of the multi-temporal data, denoising of the subsidence model, analysis of the uncertainty, and improvement in the accuracy of monitoring were studied in depth, with a view to improving monitoring performance and to promote use of the new technique in coal mining operations.

## 2. Overview of the Study Area

The area of study was the working face of the No. 1 coal mine of the Yangmei Group. The main characteristics of the mine were as follows: the strike length was 1345 m, the dip length was 226 m, the average dip angle of the coal seam was 4°, the average mining depth was 446.8 m, and the average coal thickness was 7.24 m. A half strike observation line A, and a dip observation line B, were arranged. The relative positional relationship between the working face, the observation stations, and the research area is illustrated in Figure 1.

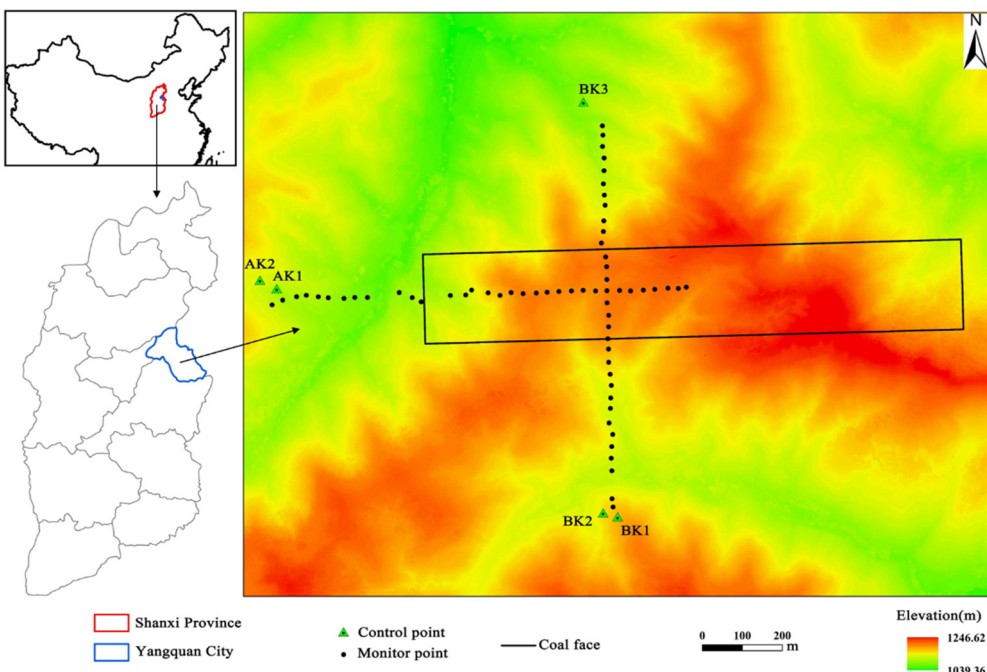

**Figure 1.** Schematic for relationship between the working face, observation stations, and study area.

### 3. Data and Methodology

*3.1. Data*

A summary of the data collection parameters is presented in Table 1. The data of 14 June 2020 have been abbreviated to 06.14 as are the other data.

**Table 1.** Time series data collection statistics.

| Number | Time | Acquisition | Result Form | Point Cloud Number | Area km² | Density per m² |
|---|---|---|---|---|---|---|
| 1 | 14 June 2020 | UAV image | Point cloud, DSM, DOM | $2.3 \times 10^8$ | 4.5 | 52 |
| 2 | 20 July 2020 | UAV image | Point cloud, DSM, DOM | $2.4 \times 10^8$ | 4.5 | 53 |
| 3 | 7 September 2020 | UAV image | Point cloud, DSM, DOM | $2.2 \times 10^8$ | 4.2 | 53 |
| 4 | 15 November 2020 | UAV image | Point cloud, DSM, DOM | $2.4 \times 10^8$ | 4.5 | 53 |
| 5 | 31 July 2021 | UAV image | Point cloud, DSM, DOM | $4.7 \times 10^8$ | 4.3 | 110 |
| 6 | 16 January 2022 | Airborne LiDAR | Point cloud | $2.5 \times 10^8$ | 4.0 | 62 |

The FEIMA D2000 quad-rotor UAV equipped with a SONY a6000 camera was used to collect the visible images and simulate the ground flight. A ground sample distance (GSD) of 4 cm/pixel was provided at a flight height of 255 m relative to the ground. Each flight was set with 80% forward overlaps and 60% side overlaps. Photoscan software was used to generate the UAV point clouds, the digital surface model (DSM) and the digital orthophoto maps (DOMs). The D-LIDAR 2000 module carried by the D2000 UAV served to collect the original LiDAR data, and the LAS-format point cloud was obtained by preprocessing. Detailed information about the camera and LiDAR module are shown in Table 2.

First, the outliers in the point cloud were removed by the statistical outlier removal (SOR) filter tool in CloudCompare software(2.10, https://www.danielgm.net/cc/). The filter performs a statistical analysis on the neighborhood of each point and calculates the average distance from it to all adjacent points. Points whose average distance were outside the standard range (defined by the global distance average plus the defined standard deviation) would be defined as outliers and removed from the data. Then, the progressive triangulation densification filtering algorithm in TerraSolid software (2019, Finland, https://terrasolid.com/) was used to automatically classify ground points, and the DEM generated from the initial filtering result was visually inspected, the corresponding misclassified

point clouds were detected through the section line, and the misclassified points were manually corrected using the classification tool to improve the initial filtering result. Finally, multi-temporal DEMs were generated by interpolation of ground point cloud. All data coordinate systems were of the WGS-84 to ensure consistency of the coordinate datum.

**Table 2.** Parameters configuration of aerial survey module and LiDAR module.

| D-CAM2000 Aerial Module | | D-LiDAR2000 LiDAR Module | |
|---|---|---|---|
| Camera | SONY a6000 | Ranging | 190 m@10%Reflectivity@100 klx<br>450 m@80%Reflectivity@0 klx |
| Effective pixels | 24.3 million | Scanning frequency | 240 kHz |
| Sensor | $23.5 \times 15.6$ mm (aps-c) | Ranging accuracy | $\pm 2$ cm |
| Focal length | 25 mm | Horizontal positioning accuracy | 0.02 m |

The average density of the original point cloud of 06.14 data is 52 per $m^2$, the average density of ground point cloud obtained by point cloud filtering is about 3.5 per $m^2$, the average spacing of point cloud is about 0.5 m, and the resolution of DEM is 0.5 m. In order to enable differential calculation of data in different periods, the DEM data of other periods were resampled to the same spatial resolution.

*3.2. Methodology*

This study integrated airborne LiDAR data and multi-temporal UAV photogrammetry data and proposed a method for monitoring mining subsidence featuring co-registration of UAV LiDAR data and photogrammetry. The process consists of four steps: first, the airborne LiDAR point cloud was used as the reference data to co-register the multi- temporal UAV dense matching point cloud, and then the performance of the co-registration was evaluated through the calculated M3C2 distance. Further, a dynamic subsidence basin for surface subsidence monitoring was constructed based on the DoD obtained from the UAV dense matching point cloud filtering and interpolation. Finally, the development process and subsidence law of the subsidence basin was studied. On this basis, we analyzed the accuracy of the dynamic subsidence basin by comparing it with the measured points in field investigation, and also quantified the uncertainty of the DoD. The flow chart of this study is shown in Figure 2.

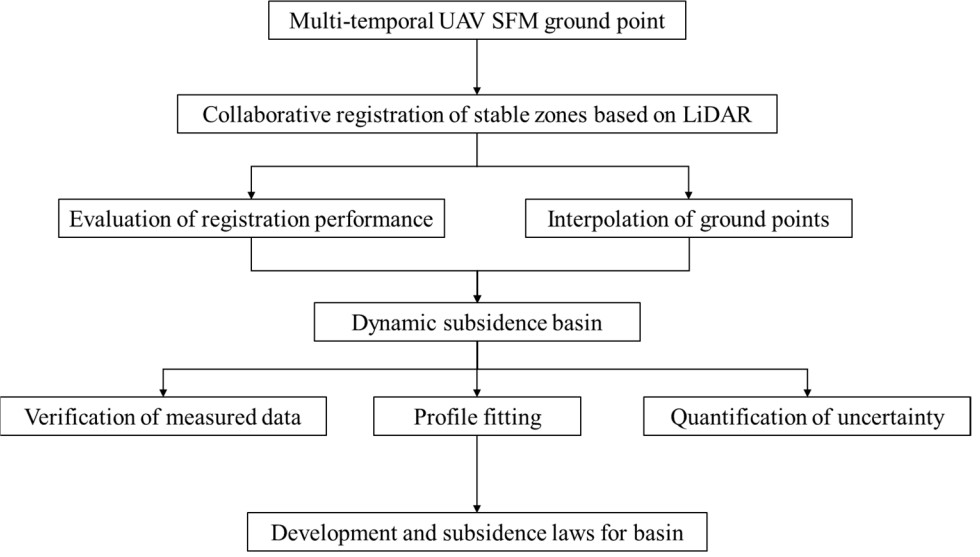

**Figure 2.** The flow chart for subsidence monitoring via UAV.

## 4. Results and Analysis

### 4.1. Co-Registration and Performance Evaluation

Taking the airborne LiDAR point cloud of 01.16 as the reference, the closest iterative point (ICP) algorithm in the CloudCompare software was used to register automatically the five-stage dense matching point cloud. Specifically, the transformed parameters calculated by the ICP algorithm on the point cloud subset of the stable region were applied to the entire point cloud. Three stable regions were selected near the control points as identified in the observation of the surface movement, as depicted in Figure 3a. These regions had less vegetation and almost no subsidence. Figure 3b represents the number of overlapping images used to calculate each pixel; also, the data obtained in the stable region would not be affected by insufficient image overlap.

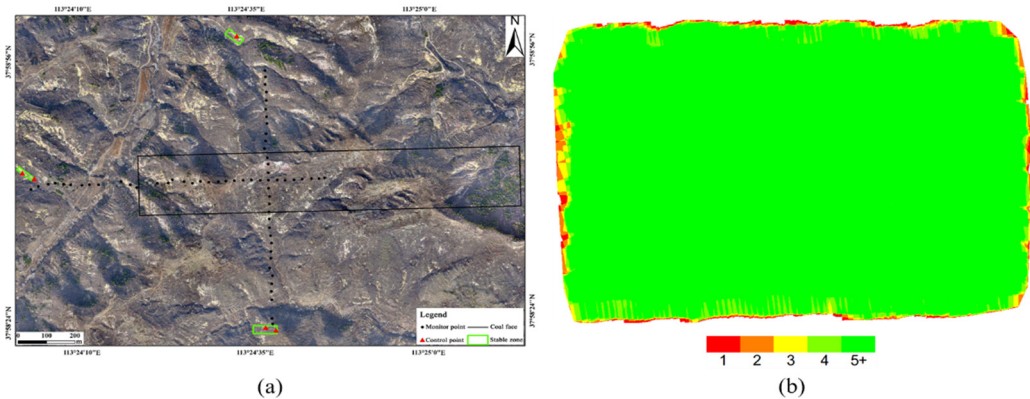

**Figure 3.** Selection of the stable region: (**a**) stable zone for collaborative registration; and (**b**) number of overlapping images computed for each pixel of the DOM.

The Multiscale Model to Model Cloud Comparison (M3C2) algorithm of the Cloud-Compare software was used to calculate the distance between the multi-temporal point clouds in the stable region after co-registration to evaluate the quality of co-registration [22]. The standard deviation of the distance was used as an indicator of the measurement precision, and the mean was regarded as a measure of accuracy of the point cloud. The results are presented in Table 3. The microtopography of the stable surface would change due to weathering or vegetation, therefore, it was difficult for the results for co-registration to reach the theoretical value of 0 m.

**Table 3.** The mean and standard deviation of the M3C2 distance in the stable region.

| Dataset | Mean (m) | Standard Deviation (m) | Duration (day) | Platform |
|---|---|---|---|---|
| 01.16/06.14 | 0.24 | 0.13 | 581 | LiDAR/UAV |
| 01.16/07.20 | 0.34 | 0.15 | 545 | LiDAR/UAV |
| 01.16/09.07 | 0.30 | 0.19 | 496 | LiDAR/UAV |
| 01.16/11.15 | 0.32 | 0.14 | 427 | LiDAR/UAV |
| 01.16/07.31 | 0.35 | 0.15 | 169 | LiDAR/UAV |

As shown in Figure 4, the calculated M3C2 histogram reveals that the distribution is Gaussian in shape but with a certain degree of positive skewness, the red bars represent the number of point clouds in each interval and the vertical black lines represent where the mean is located. Furthermore, the mean values for the data of 07.20 and 07.31 are relatively large, and the presence of dense vegetation has a certain impact on the results at these time periods. The standard deviation for the distance was between 0.13 and 0.19, indicating that the repeatability of the UAV multi-temporal photogrammetry data was good.

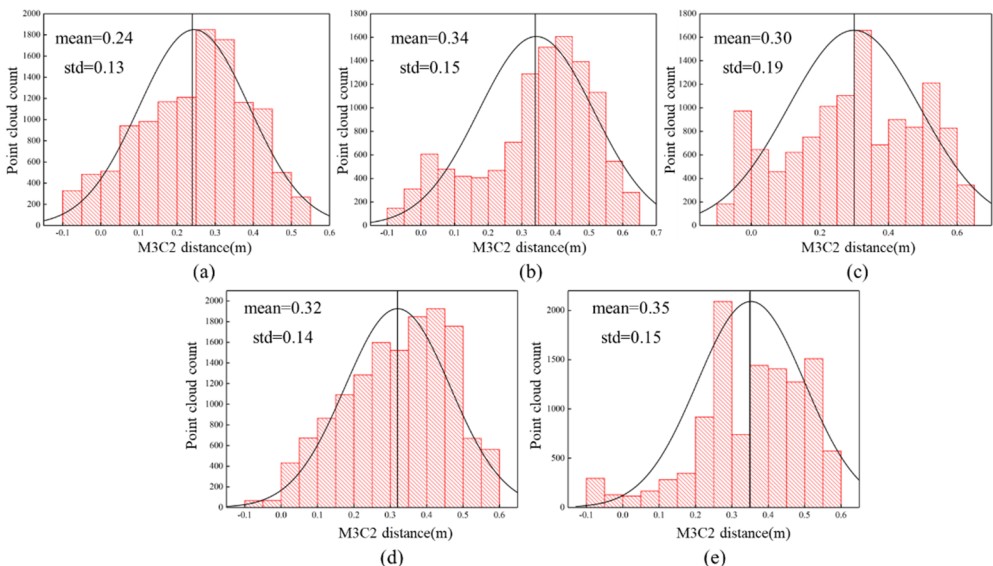

**Figure 4.** Histogram of the M3C2 distance: (**a**) LiDAR01.16−UAV06.14; (**b**) LiDAR01.16−UAV07.20; (**c**) LiDAR01.16−UAV09.07; (**d**) LiDAR01.16−UAV11.15; and (**e**) LiDAR01.16−UAV07.31.

*4.2. Construction and Verification of Accuracy for the Dynamic Subsidence Basin*

The dynamic subsidence basin was constructed using DoD analysis, as shown in Figure 5a–e is the DOM for each period, Figure 5f–i is the time series subsidence map obtained by subtracting the DEM of two adjacent periods, Figure 5j–m is the cumulative time-series subsidence map obtained by subtracting 07.20, 09.07, 11.15, and 07.31, respectively, from 06.14; the outliers are shown in the white grids. In the cumulative time series subsidence map, as the working face advanced, the range of influence of the surface expanded accordingly, the maximum subsidence value increased gradually, and the range of the subsidence basin became closer to the center of the goaf.

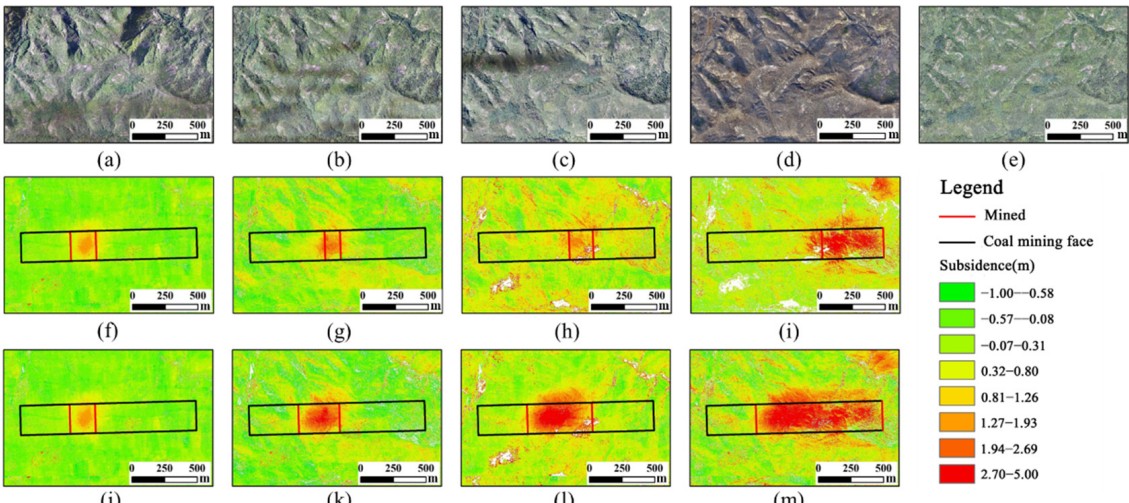

**Figure 5.** Development process for the dynamic subsidence basin at the working face: (**a**) 1−06.14; (**b**) 2−07.20; (**c**) 3−09.07; (**d**) 4−11.15; (**e**) 5−07.31; (**f**) 1−2; (**g**) 2−3; (**h**) 3−4; (**i**) 4−5; (**j**) 1−2; (**k**) 1−3; (**l**) 1−4; and (**m**) 1−5.

Based on the total measured data for the station, the accuracy for the subsidence basin was verified by comparing the difference in elevation between the measured data on the monitoring point with the difference in elevation extracted from the UAV subsidence DEM.

The mean error (ME), the mean absolute error (MAE), and the root mean square error (RMSE) may be calculated as follows:

$$ME = \frac{1}{n}\sum_{i=1}^{n}(H - h) \tag{1}$$

$$MAE = \frac{1}{n}\sum_{i=1}^{n}(|H - h|) \tag{2}$$

$$RMSE = \sqrt{\frac{1}{n}\sum_{i=1}^{n}(H - h)^2} \tag{3}$$

where $H$ and $h$ refer to the measured elevation difference and the elevation difference extracted from the subsidence DEM, respectively.

The calculated results are presented in Table 4, and where the average error was small, being in the range ±0.2 m. The RMSE was greatly affected by the abnormal value, and was generally greater than the average error and the average absolute error, most of which being between 0.2 m and 0.3 m. The minimum error was 0.17 m. Under the influence of time decoherence, the RMSE of the data with the larger time interval of 06.14 is larger, and the results of line A and line B for 07.31−06.14 are more than 0.3 m.

**Table 4.** Comparison of subsidence value error calculation between measured points and UAV extraction.

| Data Set | Average Error (m) | | Average Absolute Error (m) | | Root Mean Square Error (m) | |
|---|---|---|---|---|---|---|
| | Line A | Line B | Line A | Line B | Line A | Line B |
| 07.20−06.14 | −0.16 | −0.11 | −0.20 | 0.14 | 0.24 | 0.17 |
| 09.07−06.14 | −0.06 | −0.14 | 0.25 | 0.20 | 0.27 | 0.23 |
| 11.15−06.14 | 0.13 | −0.10 | 0.26 | 0.22 | 0.30 | 0.27 |
| 07.31−06.14 | 0.06 | −0.28 | 0.31 | 0.29 | 0.34 | 0.32 |

Taking the intersection of line A and line B as the origin, the measured subsidence was compared with the subsidence for the monitoring points in the DEM, as shown in Figure 6. The subsidence curve extracted by the subsidence DEM was close to the measured subsidence curve, and the deviation was small near the maximum subsidence value. However, many jump points in the elevation may be observed in the subsidence value extracted by the subsidence DEM.

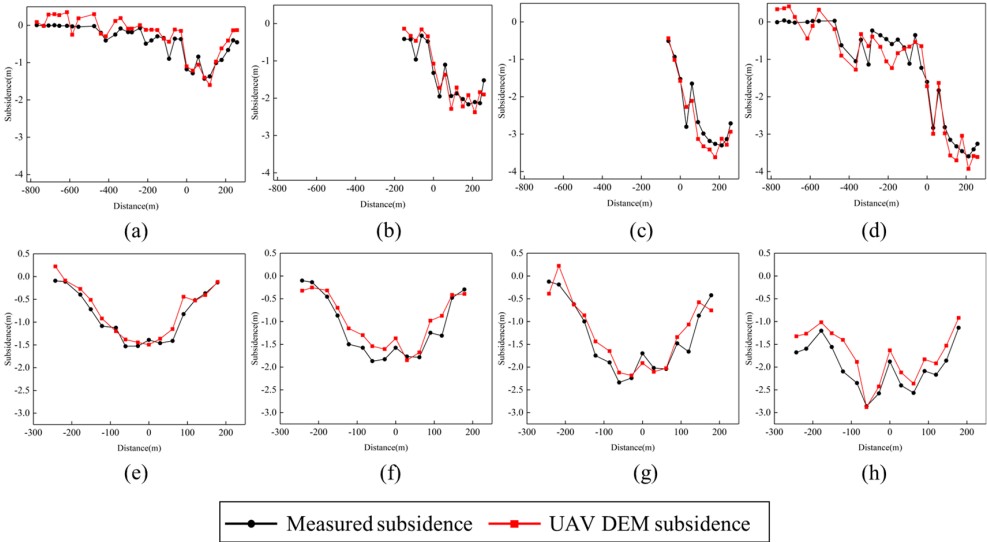

**Figure 6.** Comparison of subsidence values about the monitoring points: (**a**) line A 07.20−06.14; (**b**) line A 09.07−06.14; (**c**) line A 11.15−06.14; (**d**) line A 07.31−06.14; (**e**) line B 07.20−06.14; (**f**) line B 09.07−06.14; (**g**) line B 11.15−06.14; and (**h**) line B 07.31−06.14.

### 4.3. Analysis of the Subsidence Characteristics of the Main Section

Analysis of the main section is essential for the monitoring of mining subsidence and relies on robust line analysis at the discrete monitoring points. To this end, the section lines on the four subsidence DEMs were extracted from the main sections of the strike and dip of the subsidence basin along the working face at 0.5 m intervals to ensure that the start and end coordinates of the section lines were the same, and the values were graphed, as shown by the black scatters in Figure 7. The direction of the strike and dip profiles is from left to right and from top to bottom, with lengths of 1600 m and 910 m, respectively.

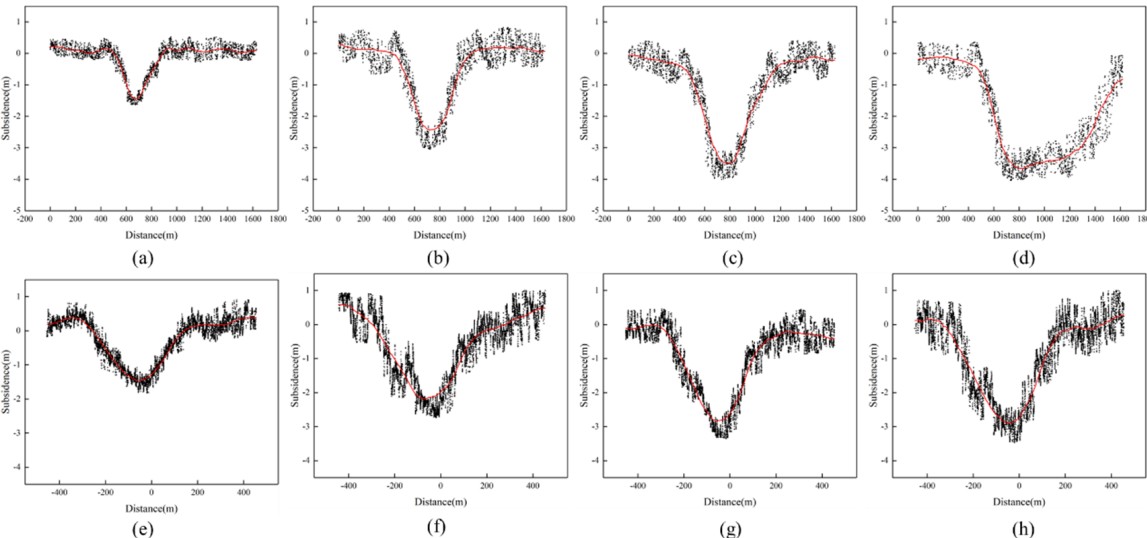

**Figure 7.** Curve fitting of subsidence and scattered points on the profile line: (**a**) strike 07.20−06.14; (**b**) strike 09.07−06.14; (**c**) strike 11.15−06.14; (**d**) strike 07.31−06.14; (**e**) dip 07.20−06.14; (**f**) dip 09.07−06.14; (**g**) dip 11.15−06.14; and (**h**) dip 07.31−06.14.

The Savitzky–Golay smoothing denoising method was used to fit the scatter data. This method is based on the use of a polynomial low-pass filter. As can be seen in Figure 7, the low-frequency signal changes slowly and the waveform is smooth, while the high-frequency signal changes quickly and abruptly.

The instability in altitude during flight leads to more jump points in the elevation in the UAV data, therefore, Savitzky–Golay smoothing was suitable for reducing the mutation noise in elevation. Finally, the time series subsidence curve for the main section was produced, as is shown by the red curves in Figure 7.

It can be seen from Figure 8 that the maximum subsidence values for the strike and the dip increased regularly, and the subsidence velocity of the last data was reduced significantly compared with that of the previous three periods; also, the subsidence value was close to the maximum value under the geological mining conditions. In the time-series strike subsidence curve, the area of maximum subsidence tended to move to the right, which was consistent with the direction of mining of the work face. The comparison between the maximum subsidence value of the fitting curve and the measured value is shown in Table 5. The relative error between the fitted and measured value was not more than 20%, and the minimum value was 0.7%. Use of UAV photogrammetry to monitor the mining subsidence can reflect comprehensively the range of influence of mining subsidence, and a high level of accuracy for monitoring the maximum subsidence value can be realized.

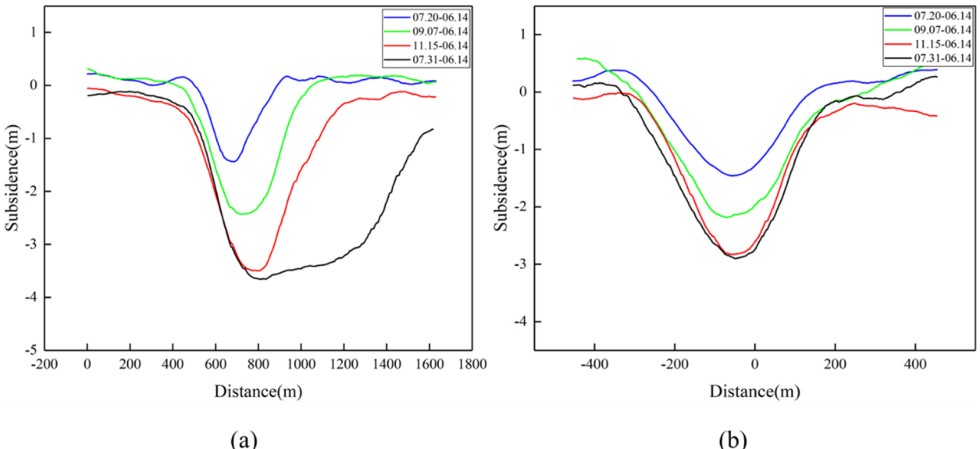

**Figure 8.** Time series subsidence curve for the main section: (**a**) strike subsidence curve fitting; and (**b**) dip subsidence curve fitting.

**Table 5.** Comparison of maximum subsidence values between measured values and curve fitting.

| Data Set | Measured Value (m) | Fitting Value (m) | Relative Error (%) |
|---|---|---|---|
| Line A 07.20-06.14 | −1.43 | −1.44 | 0.70 |
| Line A 09.07-06.14 | −2.17 | −2.43 | 11.98 |
| Line A 11.15-06.14 | −3.30 | −3.50 | 6.06 |
| Line A 07.31-06.14 | −3.60 | −3.66 | 1.67 |
| Line B 07.20-06.14 | −1.53 | −1.46 | 4.58 |
| Line B 09.07-06.14 | −1.86 | −2.19 | 17.74 |
| Line B 11.15-06.14 | −2.33 | −2.78 | 19.31 |
| Line B 07.31-06.14 | −2.86 | −2.90 | 1.40 |

Figures 7 and 8 are the comparison of the main section profiles of strike and dip extracted from the subsidence DEM in different periods, showing the dynamic development process of the subsidence curve in the form of line, which conforms to the development law of the mining subsidence curve and verifies the effectiveness of the curve fitting and subsidence monitoring methods.

*4.4. Quantification of Uncertainty in the DoD*

The uncertainty of the data must be considered to distinguish between real changes in the terrain and the noise generated by the various sources of error. The interpretation of the uncertainty in the DoD typically requires two steps:

1.　Propagating the uncertainty in an individual DEM to DoD

The uncertainty sources in the DEM generated by interpolation of the data for the ground points include the system error of the UAV, the accuracy and density of the point cloud, the filtering error, the surface composition, the sampling interval, and the interpolation method. Uncertainty is expressed as $\delta z$, and ignoring the horizontal component, the relationship between $\delta z$ and the actual elevation $Z_{Actual}$ is as follows:

$$Z_{Actual} = Z_{DEM} \pm \delta_Z \tag{4}$$

where $Z_{Actual}$ is the true elevation value, and $Z_{DEM}$ is the actual elevation value.

Use of the RMSE value based on the checkpoint data is one of the most common methods to estimate the DEM uncertainty $\delta z$. Two methods can be used to evaluate the vertical accuracy of data: comparison with field observation data and comparison with two point clouds [23]. For rigorous estimation of the error, we need to collect the complete checkpoint data for the ground, which clearly is difficult to achieve in mountainous terrain.

Therefore, the standard deviation of the M3C2 distance for evaluation in multi-temporal point cloud co-registration is used as an index of the DEM uncertainty $\delta z$.

When calculating the value of the differential DEM, the error will propagate to DoD, and the propagation error ($\delta \mu$) is determined as follows [24]:

$$\delta uDoD = t\sqrt{(\delta z_{new})^2 + (\delta z_{old})^2}$$ (5)

where $\delta uDoD$ is the propagation error, and $\delta z_{new}$ and $\delta z_{old}$ are the individual errors in the two DEMs which have been subtracted, respectively, assuming that the error in each grid is random and independent, t is the critical Student's *t*-value at a user-chosen confidence interval (i.e., $t = 1$ for 68% confidence interval) [25,26].If the spatially explicit estimation of $\delta z_{new}$ and $\delta z_{old}$ does not exist, the combined error can be calculated as a single value of the entire DoD [27].

2. Assessment of the significance of the propagated uncertainty

An assessment of the significance of the change in the uncertainty of the elevation in the DoD depends on the threshold selected and whether we discard or apply lower weights for changes in height below the minimum level of detection (minLoD). The propagation uncertainty (i.e., $\delta uDoD$) is used to define the threshold for the change in elevation or the minLoD. In the DoD data of 06.14–07.20, the standard deviation values for the M3C2 distance calculated by the airborne LiDAR data of 06.14 and 07.20 were 0.13 m and 0.15 m, respectively, such that a minLoD of 0.20 m was calculated. The change between −0.2 m and 0.2 m in the DoD was considered insignificant and discarded from the results.

The variation in the visibility of the change in elevation with the minLoD threshold in the DoD is shown in Figure 9. The original DoD with no elevation threshold is shown on the far left, and gradually the more conservative the DoD becomes, the larger the MinLoD becomes, as shown on the right. When the DEM is more uncertain, the minLoD threshold is higher, and more information is lost in the DoD for assessment of the significance during a change in elevation, and a more reliable DoD is obtained. Although UAV data may be obtained in mountainous regions and areas with dense vegetation, the DEM derived from dense point clouds has potential for uncertainty in the quantification process and for detecting change. The DoD with a minLoD threshold facilitates the detection of small changes in elevation which may be associated with errors and can reliably quantify topographic changes caused by subsidence.

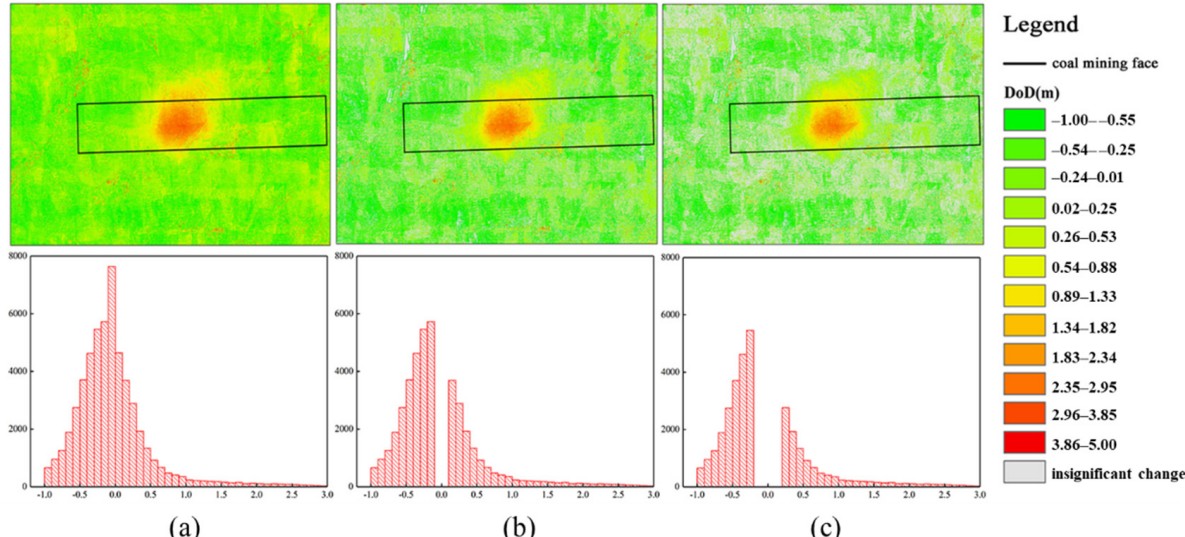

**Figure 9.** MinLoD threshold assessment of the significance for the variability in the DoD elevation: (**a**) minlod = 0.00 m; (**b**) minlod = 0.10 m; and (**c**) minlod = 0.20 m.

### 5. Conclusions

1.  Taking the standard deviation of M3C2 distance as the index, the repeatability of the SfM-UAV time-series data was evaluated via airborne LiDAR data. The results showed that the standard deviation of the M3C2 distance was between 0.14 and 0.19, which shows that the repeatability of the multi-temporal photogrammetric data of the UAV was good;

2.  The dynamic subsidence basin constructed by DoD analysis can reveal clearly the development process of surface movement in the basin. As the working face advances, the range of influence of the surface will expand correspondingly, and the maximum subsidence value will gradually increase;

3.  The RMSE of the difference in elevation between the measured monitoring points and that from the subsidence DEM extraction is mostly between 0.2 m and 0.3 m, with the highest accuracy being up to 0.17 m. The relative error between the maximum subsidence value fitted by the profile line of the main section and the measured value was less than 20%, and the minimum value was 0.7%. The accuracy of UAV subsidence DEM monitoring the maximum subsidence value is high;

4.  The DEM derived from the dense matching point cloud of the UAV has the potential to estimate the uncertainty and detect changes in elevation. The DoD with a minLoD threshold is helpful for detecting small changes in elevation that may be related to experimental errors, thus permitting us to quantify reliably topographic changes caused by subsidence.

**Author Contributions:** Conceptualization, J.L.; methodology, X.L. (Xiaoyu Liu) and X.L. (Xugang Lian); software, X.L. (Xiaoyu Liu); validation, X.L. (Xieyu Lv), B.W. and X.L. (Xugang Lian); formal analysis, J.L.; investigation, X.L. (Xiaoyu Liu) and X.L. (Xugang Lian); resources, J.L. and X.L. (Xugang Lian); data curation, X.L. (Xugang Lian); writing—original draft preparation, X.L. (Xiaoyu Liu); writing—review and editing, X.L. (Xieyu Lv) and X.L. (Xugang Lian); visualization, X.L. (Xieyu Lv); supervision, J.L. and X.L. (Xugang Lian); project administration, J.L. and X.L. (Xugang Lian); funding acquisition, J.L. and X.L. (Xugang Lian). All authors have read and agreed to the published version of the manuscript.

**Funding:** This research was funded by the Scientific Research Fund of Guizhou Provincial Education Department of China, grant number KY [2017] 097; Guizhou Province High-level Innovative Talents Project, grant number Graduation Talents [2021] 09; National Natural Science Foundation of China, grant number 42101414; National Natural Science Foundation of China, grant number 51704205; Natural Science Foundation of Shanxi Province, grant number 201901D111074.

**Institutional Review Board Statement:** Not applicable.

**Informed Consent Statement:** Not applicable.

**Data Availability Statement:** Not applicable.

**Acknowledgments:** We would like to thank the reviewers for their helpful comments.

**Conflicts of Interest:** The authors declare no conflict of interest.

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
