# Peer review of "Novel Method for Monitoring Mining Subsidence Featuring Co-Registration of UAV LiDAR Data and Photogrammetry"

_applsci, doi:10.3390/app12189374_

Round 1

Reviewer 1 Report

In this paper, authors constructed a differential digital elevation model (DEM)-based dynamic subsidence basin using time-series unmanned aerial vehicle (UAV) photogrammetry with LiDAR data, and built a differential digital elevation model (DEM)-based dynamic subsidence basin to monitor mine subsidence using this method. However, some improvements are needed before publication. My comments are as follow.

1.       First, please pay attention to the article formatting requirements and revise the location of Table 3 and Table 4;

2.       It is suggested to add the status of research on mining subsidence monitoring by domestic and foreign scholars in the introduction section and compare the advantages of the research method of this paper;

3.       It is suggested that the color band distribution of the pictures in the text be unified. In Figure 1, the high values are in red, but in Figure 3, Figure 5 and Figure 9, the high values are in green;

4.       Please explain how the stable region is determined in Figure 3 according to (b);

5.       Whether the LiDAR point cloud density data and the resolution of the constructed DEM can be provided in the text. Comparison with the DEM resolution through the average point cloud density, the average density of the filtered standard ground point cloud and the average spacing of the point cloud, in order to facilitate the assurance of point cloud data coverage in each raster;

6.       Please clarify the UAV flight altitude and scanning data accuracy for Airborne LiDAR data in the text. Since a lower UAV flight altitude can reduce the DEM errors caused by point cloud plane position errors, please clarify whether the UAV flight altitude is 255 m for both data;

7.       It is suggested that the standard deviation index be added to the accuracy verification section in section 4.2. Standard deviation measures the degree of dispersion of a set of data itself, which can be effectively compared with root mean square error, mean absolute error and mean error for analytical studies;

8.       Please add a description of the role of Figure 7 and Figure 8 in section 4.3;

9.       We suggest adding a section in the text to introduce the main methods and formulas used in the article;

10.   In Section 4.4, page 8, lines 127-128: the authors state that it is known that δ_z tends to be coherent and shows predictable spatial variation patterns , a conclusion that mostly applies to plain areas with low elevation differences, and suggest a change;

11.   In section 4.4, please simplify and integrate parts 1, 2 and 3. 1 is a general knowledge part that does not need to be specified and can be simplified by integrating it with 2 and 3;

12.       Equation (5)  in this study has a large error value. The formula is applicable to the plain areas with relatively smooth and small degree of ground undulation, and when the difference between DEM and actual value is small, the spatial change pattern can be predicted, while the ground undulation in this study area is large, the error value is large, and it is suggested to modify;

13.   Please explain how the accuracy of the DEM model itself is ensured in the study. The DEM model has spatial position offset error and non-removal of non-ground point error. Various studies have shown that a small plane error at the edge of steep hills, buildings, etc. can cause significant errors in the differential DEM, and since there are a large number of steep slopes in the study area, please clarify how to ensure the accuracy of the DEM model itself;

14.   Please explain how the threshold is determined in section 4.4, part 3, and whether a lower weight value is discarded or selected for height changes below the minimum level of detection (minLoD).

Reviewer 2 Report

This study developed a method for monitoring subsidence in mining areas using UAV LiDAR and photogrammetry. It has the great potential to achieve the higher efficiency and accuracy of mining subsidence monitoring. However, to make it easier to understand, the reviewer believes that the following comments need to be considered.

(1) In the Introduction section, more references should be included to illustrate the development of the surface subsidence monitoring technology. Meanwhile, the reviewer thinks that both satellite and airborne LiDAR are proven techniques now to obtain the dynamic and efficient data, but the photogrammetry of UAV platform depends algorithms and postprocessing techniques. How can authors ensure the applicability of the mentioned method?

(2) In the Data acquisition and processing section, please provide details of the camera and LiDAR module, such as ranging, scanning frequency, etc.

(3) To verify the data accuracy of the UAV, there should be calibrated by the GPS data of fixed checkpoints in the testing area. How are these points set?

(4) How is the outlier or the noise error processed in the point cloud?

(5) At Line 142, is there one or several monitoring points, and how is the elevation determined?

(6) How does the author address the effects of vegetation and wind speed on measurements?
